# All-to-all reconfigurability with sparse and higher-order Ising machines

Srijan Nikhar[1,2], Sidharth Kannan[1,2], Navid Anjum Aadit [1,2], Shuvro Chowdhury [1] & Kerem Y. Camsari [1] ✉

Domain-specific hardware to solve computationally hard optimization problems has generated tremendous excitement. Here, we evaluate probabilistic bit (p-bit) based Ising Machines (IM) on the 3-Regular 3-Exclusive OR Satisfiability (3R3X), as a representative hard optimization problem. We first introduce a multiplexed architecture that emulates all-to-all network functionality while maintaining highly parallelized chromatic Gibbs sampling. We implement this architecture in a single Field-Programmable Gate Array (FPGA) and show that running the adaptive parallel tempering algorithm demonstrates competitive algorithmic and prefactor advantages over alternative IMs by D-Wave, Toshiba, and Fujitsu. We also implement higher-order interactions that lead to better prefactors without changing algorithmic scaling for the XORSAT problem. Even though FPGA implementations of p-bits are still not quite as fast as the best possible greedy algorithms accelerated on Graphics Processing Units (GPU), scaled magnetic versions of p-bit IMs could lead to orders of magnitude improvements over the state of the art for generic optimization.

Ising machines and domain-specific accelerators for hard optimization problems have generated tremendous interest lately. Different implementations using various physics-inspired approaches have been implemented using a range of physical substrates (see ref. 1 for a comprehensive review). Unlike D-Wave's quantum Ising machines that operate with sparse connectivity, most recent Ising machines have emphasized all-to-all network topologies[2–5], motivated by the ease of reconfigurability. While all-to-all topology eases programming different instances of optimization problems expressed as Ising Hamiltonians, in our view, this is a temporary solution[6] since the $\mathcal{O}(n^2)$ dependence of weights will be unsustainable at large scales. The scaling problem becomes even harder if the all-to-all Hamiltonians include higher-order terms $\mathcal{O}(n^k)$, where $k$ is the locality of the Hamiltonian. As such, some form of sparsification or sparse-problem embedding seems unavoidable.

In this work, we focus on an emerging, physics-inspired solver based on probabilistic bits (p-bit), emphasizing sparse, massively parallel, and reconfigurable architectures capable of realizing higher-order interactions. p-bits are inspired by the statistical physics of interacting

particles encountered in nature (Fig. 1a) where particles form sparsely connected, asynchronous, and massively parallel networks (Fig. 1b). Sparsity enables a large degree of concurrency by allowing large parts of networks to be updated in parallel without introducing any errors, unlike densely connected networks that need to be updated serially[7,8]. However, sparse and fixed topologies cannot be easily reconfigured; they lack the flexibility provided by all-to-all architectures. We solve this problem by introducing a reconfigurable architecture while holding on to the massive parallelism of sparse networks. Our central idea is to introduce a master graph architecture that can multiplex different connectivity and phase-shifted (colored) clocks for a given p-bit to enable reconfigurability and parallelism. We also introduce the first implementation of third-order interactions in a large-scale FPGA-based p-computer. We implement both of these concepts in field programmable gate arrays (FPGA) and compare our implementation on a recently introduced combinatorial optimization challenge, namely the XORSAT problem[9,10] (Fig. 1c). The variant of the XORSAT problem, the challenge is based on, is known to have a polynomial time algorithm through Gaussian elimination, with $\mathcal{O}(n^3)$ complexity. Strikingly,

[1]Department of Electrical and Computer Engineering, University of California, Santa Barbara, Santa Barbara, CA 93106, USA. [2]These authors contributed equally: Srijan Nikhar, Sidharth Kannan, Navid Anjum Aadit. ✉e-mail: camsari@ece.ucsb.edu

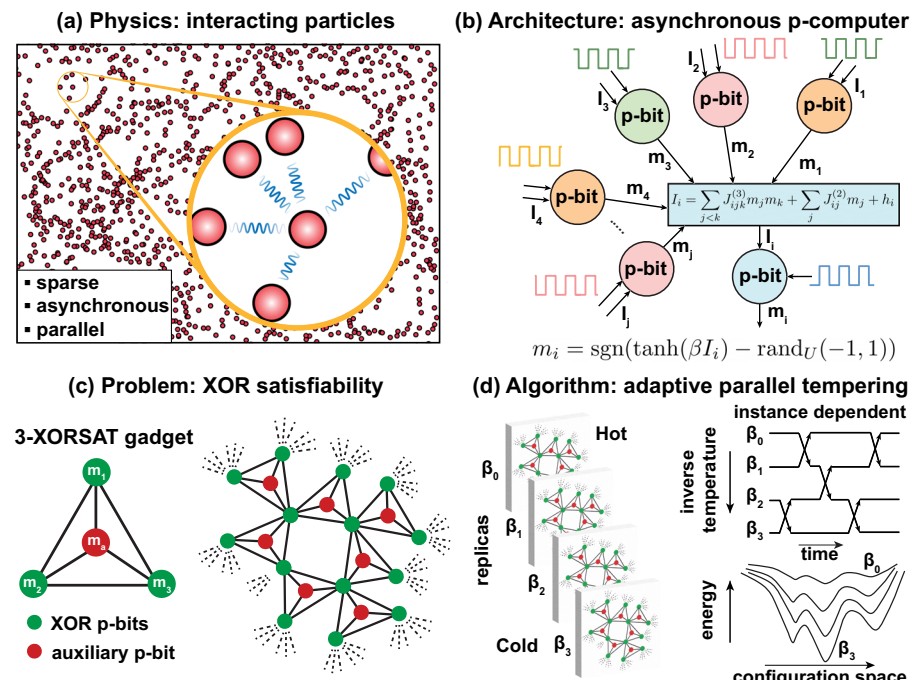

**Fig. 1 | Physics-inspired probabilistic computing. a** Many-body physics of interacting particles have local connectivity (sparse), asynchronous dynamics (clockless), and massive parallelism. **b** Asynchronous p-computers take inspiration from the physics of (**a**). Each p-bit asynchronously (shown with phase-shifted clocks) receives input from local neighbors followed by probabilistic activation. **c** The 3-regular 3-XORSAT (3R3X) problem[9,10] under study. **d** We use the adaptive parallel tempering (APT) algorithm[11,12] on FPGA-based p-computers to solve the 3R3X problem. APT uses replicas of the original network, operated at different computational temperatures where neighboring replicas swap their states based on a Metropolis criterion at regular intervals.

despite this ease, all stochastic local search solvers seem to exhibit exponential time to solution in the problem size in XORSAT, failing to distinguish the underlying structure from that of a more general and harder $k$-SAT ($k > 2$) problem. The real relevance of XORSAT comes from existing and carefully collected performance data of leading Ising machines, allowing a comparison of scaling exponents and prefactors among different approaches to solve it. Given the broad applicability and problem-agnostic nature of Ising machines, significant improvements in prefactors and/or algorithmic exponents would undoubtedly carry over to more general SAT problems.

Our FPGA-based p-computer uses a sophisticated heuristic algorithm called adaptive parallel tempering (APT)[11,12]. PT (Fig. 1d) maintains replicas of a given network at different computational temperatures, which exchange states at regular intervals. Unlike simulated annealing, PT never permanently gets stuck in local minima. Variations of PT are considered to be the most powerful heuristic algorithms of unstructured optimization problems[13]. The "adaptive" version of PT we adopt here uses a problem-dependent preprocessing step in order to identify the optimum number of replicas and temperature profiles, avoiding bottlenecks during replica swaps.

With reasonable assumptions and experimental demonstrations, we show that FPGA-based p-computers running the APT algorithm exhibit extremely competitive hardware and algorithmic scaling compared to leading IMs[4,14–17]. We demonstrate that going from a 2-body to a 3-body realization of the XORSAT problem results in a significant prefactor advantage on the time to solution and halves the number of p-bits but results in no improvement to scaling, settling a conjecture posed by Kowalsky et al.[10]. Moreover, projections based on nanodevices such as stochastic magnetic tunnel junctions (sMTJ) show that scaled p-computers could deliver unprecedented advantages in hard optimization without suffering from reconfigurability issues.

We organize the rest of the paper as follows: we first discuss the background of p-computing, followed by our multiplexed sparse network that emulates all-to-all reconfigurability on quadratized

instances, with experimental results establishing how our multiplexed architecture retains its massive parallelism without any degradation in solution quality. We then discuss how the massive parallelism provides an $\mathcal{O}(n)$ scaling advantage over serial all-to-all solvers. We extend this discussion to the multiplexed hardware implementation of third-order interactions in our digital p-computers and show how such an architecture-based scaling advantage extends to higher-order p-computers. Finally, we benchmark the XORSAT challenge in our digital p-computers and compare the performance of our second and third-order implementations against those of other leading Ising Machines.

## Results

### Background on p-bits and XORSAT
The dynamics of p-computers can be described as a discrete Markov Chain Monte Carlo (MCMC) algorithm, known as Gibbs sampling or Glauber dynamics[18]. For each p-bit in the network, we have:

$$I_i = \sum_j J_{ij} m_j + h_i \tag{1}$$

$$m_i = \mathrm{sgn}\left[\tanh(\beta I_i) - \mathrm{rand}_U(-1, 1)\right] \tag{2}$$

where $m_i \in \{-1, +1\}$ and $\mathrm{rand}_U(-1, 1)$ is a uniform random number that lies in the interval $[-1, 1]$. $\{J_{ij}\}$ and $\{h_i\}$ correspond to the weights and biases, respectively. $\beta$ is the inverse temperature. After a sufficient number of iterations, Eq. (2) approximates the Boltzmann distribution[8] given by:

$$p(\{m\}) = \frac{1}{Z} \exp[-\beta E(\{m\})] \tag{3}$$

$$E(\{m\}) = -\sum_{i<j} J_{ij} m_i m_j - \sum_i h_i m_i \tag{4}$$

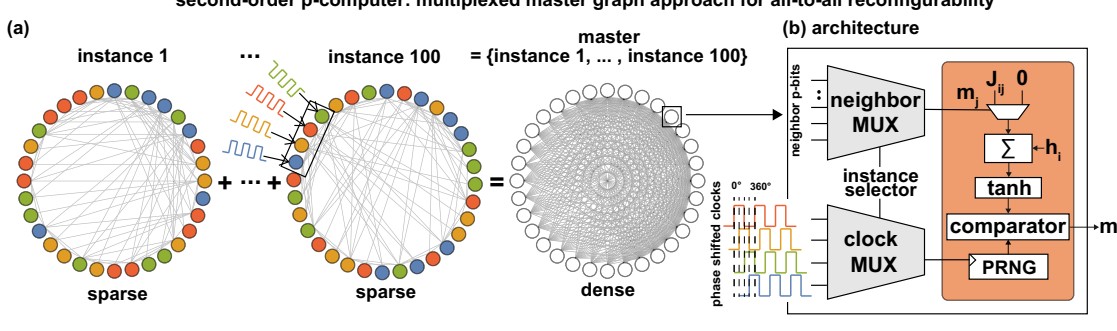

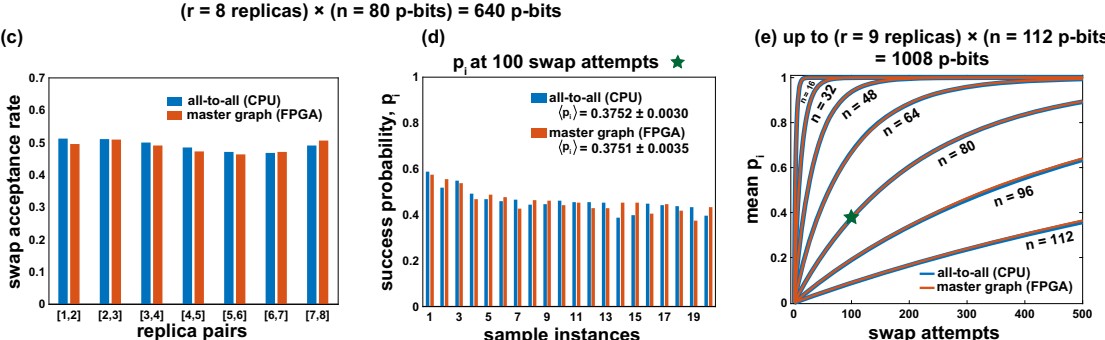

**Fig. 2 | Multiplexed all-to-all reconfigurable master graph approach. a** For each problem size ($n$), multiple graph-colored sparse instances of the 3R3X problem are combined to form a dense master graph. **b** In our architecture, neighbors, and clocks for each p-bit are multiplexed using an instance selector. **c** Pair-wise swap acceptance rates show roughly equal probability in both the master graph (FPGA) and all-to-all graph (CPU), obtained from APT across 8 replicas for $n = 80$. **d** Twenty instances with the highest success probabilities ($p_i$) are shown for $n = 80$. All $p_i$ values are computed from 1000 independent runs. **e** Mean $p_i$ as a function of swap attempts at varying $n$. **c**–**e** Establish equivalence between our master graph approach (FPGA) and all-to-all graph (CPU).

where $Z$ is the partition function. Generalizations of the Ising Hamiltonian (Eq. (4)) beyond quadratic terms are possible and have been implemented previously[19–26]. One such generalization that we implement in our p-computers is the inclusion of third-order interactions implemented by modifying Eq. (1) to:

$$I_i = \sum_{j<k} J^{(3)}_{ijk} m_j m_k + \sum_j J^{(2)}_{ij} m_j + h_i \qquad (5)$$

while keeping Eq. (2) the same. Here $\{J^{(3)}_{ijk}\}$ models the third-order interaction, whereas $\{J^{(2)}_{ij}\}$ models the second-order interaction.

Eqs. (1, 2), and (5) are extremely general and can be used to solve a wide variety of problems in optimization, sampling (for energy-based machine learning) and quantum simulation through quantum Monte Carlo[27] and variational quantum machine learning algorithms[28]. Here, we focus on the 3R3X problem, which encodes clauses of the type $C_k = (x_i \oplus x_j \oplus x_k)$, typically mapped to quadratic Ising energies via auxiliary variables (Fig. 1c), however, in this paper, we also implement a native third-order mapping without any auxiliary variables. This is a benchmark problem with a golf-course-like energy landscape that is difficult for heuristic Monte Carlo solvers to tackle[29] even though it admits a polynomial time algorithm through its relation to a system of linear equations modulo 2, solvable by Gaussian elimination[17]. Nonetheless, the problem resembles hard generic SAT problems, and slight variations of it cannot be solved by Gaussian elimination and its main value is to serve as a benchmark for hard optimization problems and Ising solvers.

**Adaptive parallel tempering**
Our p-computers run an adaptive version of the parallel tempering (APT) algorithm, in which multiple replicas of the same p-bit network are run at different temperatures in parallel (according to Eqs. (2–5))

and then periodically swapped with swap probability defined by the Metropolis criterion:

$$P_{sw} = \min(1, e^{\Delta E \Delta \beta}) \qquad (6)$$

where $P_{sw}$ is the probability of swapping the temperatures of 2 adjacent replicas, $\Delta E = E_{i+1} - E_i$ and $\Delta \beta = \beta_{i+1} - \beta_i$ are the differences in the energies and inverse temperatures of two adjacent replicas $i$ and $i+1$ (with $\beta_i < \beta_{i+1}$). This swapping enables the solver to avoid getting trapped in local minima, as high-temperature replicas explore the whole energy landscape while lower-temperature replicas exploit the most promising paths through it. The APT algorithm determines a temperature profile specific to each problem, as described in Algorithm 1. It was, however, found that the temperature profiles generated using the APT preprocessing algorithm lead only to very slight variations over all 100 XORSAT instances of a certain problem size. Therefore, for a given problem size, a fixed temperature profile generated using a randomly chosen instance of the corresponding problem size was used in all our experiments.

**All-to-all reconfigurability via sparse network multiplexing**
We define all-to-all reconfigurability as the ability to solve multiple instances of a given problem using the same hardware. We implement a dense master graph architecture on both second-order and third-order problems that can be multiplexed to access different instances of a sparse combinatorial optimization problem. While our approach cannot fully emulate all-to-all reconfigurability on natively dense problems, it has previously been shown how such problems can efficiently be sparsified while leveraging a large degree of parallelism[7]. Concretely, our approach is based on combining all 100 sparse instances of the 3R3X problem (as defined in the XORSAT challenge) into a single

master graph that allows activating one instance at a time (Fig. 2). In general, a master graph can be defined as a complete graph that can house different instances of a sparsified optimization problem, either statically multiplex (as in this paper) or dynamically reconfigured (to activate any desired instance). As we discuss in the following sections, the master graph approach must also multiplex the colors of a given node, which selects phase-shifted clocks for sparse p-bit networks that are updated in large parallel blocks.

One might initially assume that if the number of instances embedded to the master graph increases, the network topology would naturally evolve into a static all-to-all network. However, this is not the case. The idea of the master graph is about exploiting the sparsity of individual instances. Even if the master graph resembles an all-to-all network as the number of instances increases, the network always multiplexes a sparse instance and is never fully active. In all-to-all topologies, because the number of neighbors can reach the size of the network, the synaptic adders must increase in size (with complexity $\mathcal{O}(n^2)$) whereas in our approach these adders are always bounded by the number of neighbors, $k$ (with complexity $\mathcal{O}(kn)$). Moreover, the sparsity of the instances allows us to use graph coloring techniques enabling massive parallelism. This makes our approach fundamentally different from a static all-to-all network.

Choosing instances one at a time requires the master graph nodes to be reconfigurable to multiplex-changing neighbors over different instances. (Fig. 2a). Therefore, all potential neighbors of a given p-bit over different instances need to be known previously to be multiplexed using a neighbor multiplexer (Fig. 2b).

Our p-computer employs graph-colored[30] Gibbs sampling to achieve massive parallelism[7] by updating blocks of unconnected p-bits at the same time, using Eqs. (1), (2), and (5). The 3R3X instances are naturally sparse, and graph coloring their graph representation requires a maximum of 6 colors. Since only one instance is selected at a time, the master graph needs only 6 phase-shifted clocks. However, the color of a particular p-bit on the master graph can vary across instances and needs to be multiplexed as well (Fig. 2b). The phase-shifted clock signals are multiplexed using the instance selection signal before being supplied to the p-bits. In our master graph implementation, all possible weights for a given p-bit are not stored on-chip (inside the FPGA), but they are simply reprogrammed into the FPGA along with the instance selection signal from a CPU (note that this does not require any resynthesis). For a truly standalone master graph architecture, the weights can be stored in on-chip memory and loaded to solve a given instance.

For the weights ($J_{ij}$) that are modulated by the computational inverse temperature $\beta$, we have chosen a fixed-point precision: 1 bit for the sign, 6 bits for the integer part, and 6 bits for the fraction part, i.e., s{6}{6}. This optimal bit-precision was carefully chosen by comparing the master graph results against the CPU as shown in Fig. 2c–e. All our FPGA results in this work have been implemented on a single Xilinx Alveo U250 FPGA. Each phase-shifted p-bit clock is a 15 MHz clock generated in the FPGA. The random numbers required for emulating Eq. (2) are obtained using Xoshiro[31] as the pseudo-random number generator.

We performed a statistical analysis using the APT algorithm described in Algorithm 1 on 3R3X instances to verify master graph implementations on FPGA and compared our results with the all-to-all CPU implementation. It was ensured that both CPU and FPGA used the same APT parameters for our experiments. The temperature profiles and the number of replicas for each problem size were determined using the APT preprocessing step described as part of Algorithm 1. The parameters used for preprocessing were set to $\alpha = 1, \beta_0 = 1.0, \sigma_{min} = 0.5, N_{chains} = 100, L = 2000$. Each replica used for this experiment is an individual master graph that includes all 100 instances multiplexed together. For each problem size, all the replicas

were fit in the same FPGA synthesis, and the clock multiplexers were shared between them.

In all our experiments, 100 Monte Carlo (MC) sweeps were performed between 2 successive swap attempts. In this work, we define a sweep as when all the p-bits in the network (across all replicas) have attempted to flip once, as these occur in parallel in our hardware. For every problem size, APT was run on each of the 100 instances for 1000 runs, allowing us to obtain a success probability, $p_i$, for each instance. Since the ground state energy of these problems is known, before every swap attempt, the energy of every replica is calculated and compared against the ground state. The run is considered successful if any one of the replicas reaches the ground state, after which the run is terminated, and the number of swap attempts required to reach the ground state is noted. Note that this measurement approach allows an estimation of $p_i$ serving as validation experiments; however, we will use $p_i$ to estimate actual time to solution metrics later.

**Algorithm 1.**

| | |
|---|---|
| **Input:** | Weights, biases, number of swaps, sweeps per swap, colormap, step rate, initial temperature, energy variance tolerance, number of chains, sweeps per chain |
| **Output:** | State corresponding to minimum energy, $m_{opt}$ |

1 **Function** p-computer (*weights, biases, colormap, temp.*):
2 **for** *each color in the colormap* **do**
3 **for** *each p-bit in the color* **do**
4 solve Eq. 1 and Eq. 2

5 $t \leftarrow 0$
6 initialize all parallel chains to random states
7 **while** *energy variance is greater than tolerance* **do**
8 **for** *each chain in parallel* **do**
9 **for** *each sweep* **do**
10 sample p-bit states from p-computer
11 compute energy of the chain
12 Compute energy variance for the chain
13 save the p-bit states
14 compute mean energy variance of chains, $\sigma_E$
15 $\beta_{t+1} \leftarrow \beta_t + \frac{\alpha}{\sigma_E}, t \leftarrow t + 1$

16 initialize all replicas to random states
17 **for** *each swap attempt* **do**
18 **if** *it is an even-numbered swap attempt* **then**
19 choose (even, odd) sequential pairs
20 **else**
21 choose (odd, even) sequential pairs
22 **for** *each replica in parallel do* **do**
23 **for** *each sweep* **do**
24 sample p-bit states from p-computer
25 compute the energy of the replica
26 **for** *each sequential pair of replicas* **do**
27 propose a swap
28 **if** *accepted* **then**
29 swap the p-bit states between the replicas

30 **return** p-bit states for the replica with the minimum energy

## Adaptive parallel tempering with p-computers

Figure 2c shows pairwise swap acceptance probabilities for an 8 replica, 640 p-bit system corresponding to $n = 80$. Roughly uniform replica swap probabilities are observed, indicative of how the APT

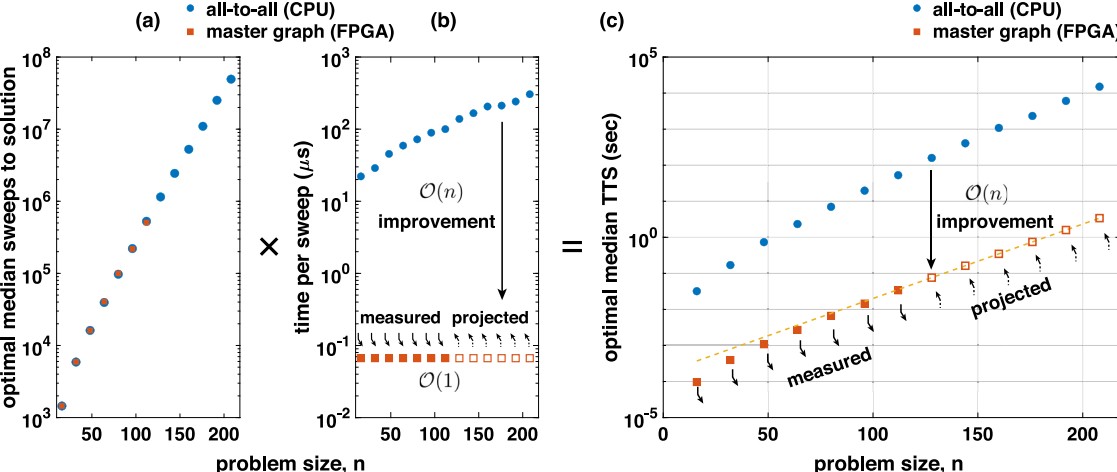

**Fig. 3 | Performance comparison between CPU and FPGA implementations of second-order p-computers. a** The algorithmic complexity of the 3R3X problem as a function of Monte Carlo (MC) sweeps to solution, independent of all-to-all (CPU) or master graph (FPGA) implementation. **b** The average time to complete one MC sweep is shown for both CPU and FPGA. For CPU, we observe an $\mathcal{O}(n)$ dependence as in[10]. For the master graph, we observe an $\mathcal{O}(1)$ dependence. **c** Multiplying (**a**) with (**b**) yields time to solution (TTS), preserving the $\mathcal{O}(n)$ improvement over the CPU (see ref. 27 for a similar analysis).

algorithm creates efficient temperature profiles, similar to the PT variant used for DAU[10].

Figure 2d compares success probabilities over sample instances between the all-to-all (CPU) and master graph (FPGA) approach for $n = 80$. Over 100 instances, the mean success probabilities between CPU and FPGA match up to 3 decimal places (errors calculated from maximum deviations at 95% confidence intervals). Finally, Fig. 2e shows mean success probabilities over all instances as a function of swap attempts between the CPU and the FPGA with excellent agreement.

Overall, the results in Fig. 2c–e establish the equivalence of our master graph approach with the all-to-all topology in CPU and verify our master graph implementation in the FPGA that achieves reconfigurability without sacrificing the massive parallelism (as we later show in Fig. 3b where the time to sample a full network stays constant across increasing problem sizes).

## Architecture-enabled scaling advantage of p-computers

Figure 3a shows the algorithmic complexity of the 3R3X problem using the APT algorithm, independent of the all-to-all (CPU) and master graph (FPGA) implementations. While Fig. 3 presents the results produced on the second-order implementation, this analysis is also extended to the third-order problems in the following sections. We report the optimal median sweeps to solutions for different instance sizes running APT to solve 100 instances per size with 1000 runs per instance. For the all-to-all (CPU) approach, we have data for all the problem sizes; however, for the master graph (FPGA) approach, we have data points up to 112 bits, which is the largest we could fit on our single FPGA. The detailed methodology of the optimal median sweeps computation is discussed in the "Methods" Section. The CPU and the FPGA data fall on top of each other, showing hardware independence and the raw algorithmic complexity despite the significant differences in the CPU implementations using float64 precision and more sophisticated RNGs (Mersenne) compared to Xoshiro. In Fig. 3b, we report the average time required to compute each sweep for both CPU and FPGA. The CPU's computation time escalates with the problem size showing $\mathcal{O}(n)$ scaling despite using the same optimized graph-coloring algorithm for block updates. Conversely, the FPGA's computation time remains constant since all the replicas and color blocks operate in parallel. This parallelism enables a linear increase in flips per second, resulting in an $\mathcal{O}(n)$ improvement for the FPGA over the CPU due to our architecture. Multiplying the optimal median sweeps from

Fig. 3a by the average time per sweep from Fig. 3b, we calculate the optimal median time to solution (Eq. (7)) in seconds in Fig. 3c. These architectural benefits are not confined to second-order implementations. In subsequent sections, we demonstrate how similar advantages were also realized in higher-order p-computers.

## Higher-order interactions

The three spin clauses of the 3R3X instances make them suitable to be represented by cubic Ising interactions. In cubic form, XORSAT problem instances need only one spin per SAT variable, without any auxiliary spins[9]. This halves the total number of the p-bits required to represent a particular problem. The details of this reduction are mentioned in the "Methods" section. To evaluate the performance of the 3R3X instances in their native third-order form, we extended the master-graph implementation discussed earlier to implement a reconfigurable third-order p-computer. The following sections discuss the details of hardware implementation for our third-order p-computer.

To preserve the massive parallelism for third-order problems, we need to extend the idea of graph-colored Gibbs sampling. Higher-order Ising systems can be represented as hypergraphs, which model multi-spin interactions as nodes connected in $n$-tuples. As in the two-body case, the XORSAT hypergraphs are sparse, with each spin being in at most three clauses, enabling a massively parallel architecture using hypergraph-colored p-bits. Extending from quadratic interactions, we enforce the strong hypergraph coloring constraint, in which two nodes that share a hyper-edge must be different colors. This creates the restriction that even in a multi-spin interaction, no two interacting nodes can update in parallel. To produce a strong hypergraph coloring, it is sufficient to color the two-body clique graph, which, in the case of the XORSAT problem, is equivalent to the two-body form of the problem, with all auxiliary spins removed. Thus, the same color assignments from the two-body problem instances can be used here. In Fig. 4, we demonstrate that such a coloring still converges to the Boltzmann distribution, while the weaker restriction that each clause must contain ≥2 colors produces a different stationary distribution.

The hardware architecture for modeling third-order Ising interactions is similar to the one used for quadratic Hamiltonians[7], except for a slight modification in the multiply-accumulate (MAC) unit to accommodate the third-order terms of Eq. (5). Because the FPGA employs binary spins, only the terms where both $m_j$ and $m_k$ are equal to

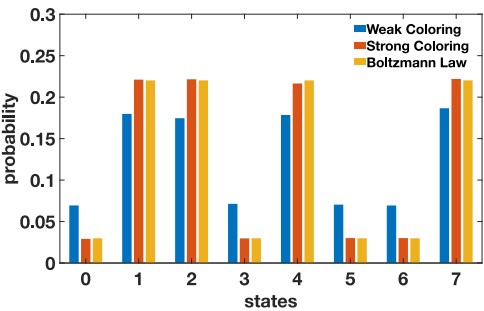

**Fig. 4 | Validating hypergraph coloring of fully connected XORSAT clause.**
Comparison among strong hypergraph coloring, weak hypergraph coloring, and Boltzmann Law is shown using $10^5$ samples. In a weak coloring, where two p-bits in the same clause can be the same color, the network does not reach the Boltzmann distribution.

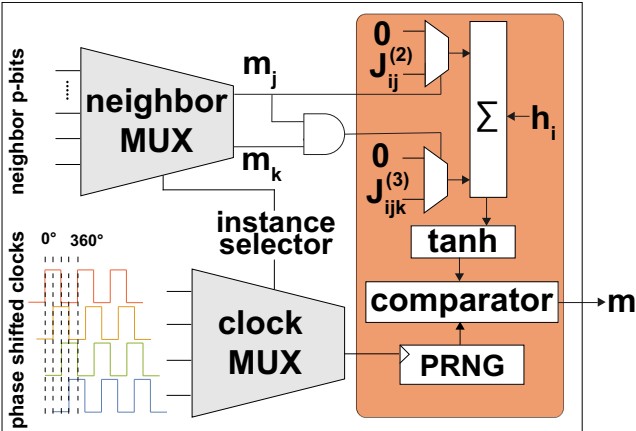

**Fig. 5 | Hardware architecture for higher order master graph.** Weights are selected using two multiplexers, one identical to the weight selector in the two-body design and the other controlled by two neighboring spins passed through an AND gate. The binary nature of p-bits greatly simplifies the higher-order interactions, avoiding multiplications.

1 contribute to Eq. (5). This scenario can be efficiently handled using a multiplexer that directs the weights to the accumulator exclusively when both associated spins are 1 (Fig. 5). Such an implementation is highly general and can be extended to incorporate arbitrarily higher-order interaction terms. Thanks to the binary nature of the p-bits, the hardware complexity to encode higher-order interactions only grows logarithmically.

The third-order master-graph implementation also builds upon the approach similar to the second-order implementation. For every p-bit, neighbors exist in pairs and must be multiplexed accordingly. We also established the equivalence between the all-to-all CPU and reconfigurable FPGA implementations using the same experiments used for the second-order implementation as shown in Fig. 6a–c. The advantages of massive parallelism are achieved by clock multiplexing using the aforementioned hypergraph coloring procedure. Thus, the hardware implementation for the third-order p-computer also gives an $\mathcal{O}(n)$ advantage over the CPU implementation. This is illustrated in Fig. 6d–f.

## p-computer results on the XORSAT challenge

Next, we present results using p-computers on the XORSAT challenge (Fig. 7), established by Kowalsky et al.[10]. We report measured data for the FPGA-based probabilistic computer's optimal median time to solution across a range of problem sizes from 16 to 112 for second-order implementation, whereas from 16 to 160 for third-order implementation, with projected performance for the remaining sizes up to 208 based on the CPU data.

The optimal median time to solution is defined as[10]:

$$\langle \text{TTS} \rangle_q = \min_{t_f} \left\langle t_f \frac{\ln(1 - 0.99)}{\ln\left[1 - p_i(t_f)\right]} \right\rangle_q \frac{1}{f_p(n)} \tag{7}$$

where $q$ represents the median ($q = 0.5$), the first ($q = 0.25$) and the second quartiles ($q = 0.75$) of TTS over the instances. $f_p(n)$ is a parallelization factor that counts how many runs of the same instance can be executed in parallel in a single solver. Even though we could fit multiple master graphs at lower $n$, we used $f_p = 1$ for the FPGA-based p-computer at all $n$. Figure 7 focuses on the median optimal TTS, i.e., $\langle \text{TTS} \rangle_{q=0.5}$ for all solvers, and Table 1 reports the scaling exponent $\gamma$ and the prefactor $\eta$ for each quartile. A detailed description of TTS calculation is mentioned in the "Methods" section.

The optimal median TTS for each solver is modeled against the size of the 3R3X instance $n$ by the relationship[10]:

$$\langle \text{TTS} \rangle_q \sim 10^{\gamma n + \eta} \tag{8}$$

Our FPGA-based p-computer's performance is reflected by a consistent slope of 0.0206, as a result of the constant average sweep time of the FPGA irrespective of problem size discussed in Fig. 3. It is interesting to note that the instance-dependent adaptive PT algorithm running on our FPGA-based p-computer shows improvements in both $\gamma$ and $\eta$ over the standard PT results reported in[10], nearly matching the performance of the PT-based DAU solver[5]. The reason for this improvement is the highly efficient graph-colored architecture[7] in our FPGA, where the entire network is updated in one clock cycle ($T_{\text{clk}} = 66.67$ ns). Without this architecture-enabled scaling advantage, the slopes with our APT and the PT[10] are roughly the same; however, the APT uses fewer replicas at all sizes, e.g., 11 replicas at $n = 208$, as opposed to 32 in PT. Note that we have only used a single instance at a given size to create the temperature profiles, and further improvements are possible with hyperparameter optimization on different instances. Kowalsky et al. originally grouped the solvers into three groups based on scaling: (i) SATonGPU and DAU, (ii) SBM, PT, and MEM, (iii) DWA where only two of the solvers represent dedicated hardware: DWA and DAU. The second-order p-computer (FPGA) shows a close performance to the leading group of solvers in (i) while outperforming (ii) and (iii), as seen in Table 1 and Fig. 7a. Moreover, our third-order implementation outperforms every solver except for SATonGPU, as illustrated through Fig. 6 and the related discussion in the following section.

Figure 7b illustrates the performance of our third-order p-computer with other solvers. Similar to Fig. 7, this figure is also adapted from ref. 10. To facilitate the comparison with other solvers, the $x$-axis represents problem sizes in terms of second-order instance sizes as in ref. 10. Therefore, all third-order instance sizes are scaled by a factor of 2. This is also followed in all references to problem size ($n$) for third-order instances in this manuscript.

Compared with the benchmarks for the second-order p-computer in Fig. 7a, a performance improvement is clearly evident in terms of the prefactor (Table 1). The third-order p-computer (FPGA) outperforms the DAU with this prefactor advantage. However, a slope of 0.0201 matches that of the second-order implementation for up to 3 decimal places. Thus, even though a prefactor advantage is seen in the third-order form, no significant improvement in scaling is observed. Even though the APT algorithm is not purely a local solver, the lack of algorithmic improvement may indicate the scaling is near-optimal, as discussed in ref. 17. At first sight, the quadratization of the natively third-order energy landscape may suggest it introduces severe difficulties for a Monte Carlo algorithm.

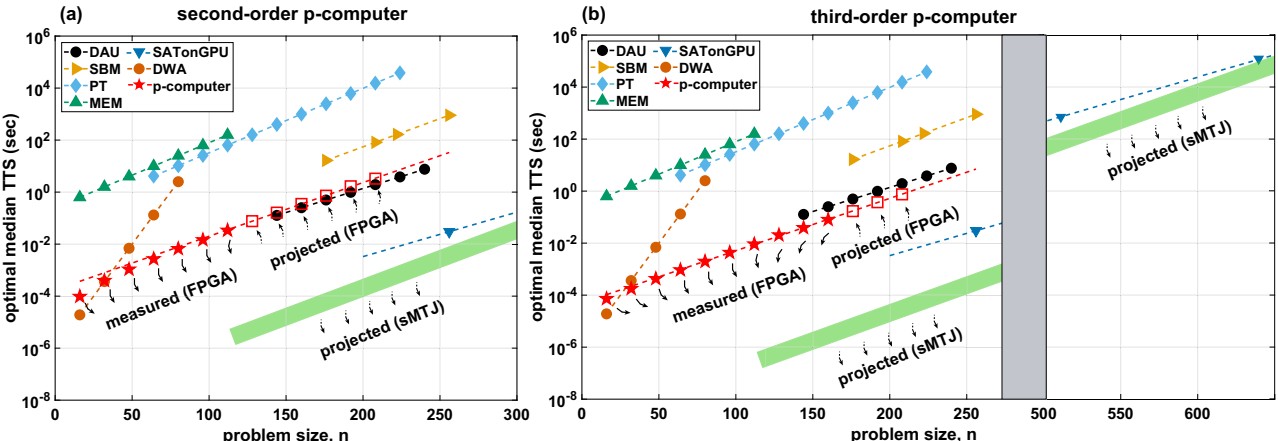

**Fig. 6 | Performance comparison between CPU and FPGA implementations of third-order p-computers. a** A roughly equal replica swap probability for 5 replicas of problem size $n = 128$ indicates a good temperature profile generated during APT. **b** The performance comparison of CPU with FPGA in terms of success probability of individual instances after 250 swap attempts and **c** across multiple problem sizes in the range of 0–250 swap attempts indicate a clear overlap between the CPU and the FPGA. **d** This overlap in algorithmic performance is visible in optimal median sweeps-to-solution as well. **e** An $\mathcal{O}(n)$ performance advantage over CPU is observed in sweep times, which is translated to **f** an overall performance benefit in the FPGA implementation of third-order p-computer.

**Fig. 7 | Optimal median time to solution (TTS).** For **a** second and **b** third-order p-computers, TTS for quartile $q = 0.5$ is shown as a function of problem size, $n$, for different state-of-the-art solvers. We adapt the figure data from ref. [10] for other solvers and add the results from the p-computer onto it. The solid red stars represent measured data for the FPGA-based p-computer. Hollow red squares indicate FPGA projections based on the CPU data from Fig. 3. Error bars obtained from 95% confidence intervals are smaller than the size of the markers and omitted. The green rectangles represent projections for stochastic Magnetic Tunnel Junction (sMTJ)-based p-computers assuming 10−50 replicas fitted in a monolithic chip with 1 million sMTJs, where each sMTJ is assumed to fluctuate with $\tau = 1$ ns, yielding 1 million flips per nanosecond for the entire network and 1 ns of sweep time at all sizes. (see the text).

**Table 1 | Scaling and prefactor fitting parameters**

| Solver | $\gamma$ | | | $\eta$ | | |
|---|---|---|---|---|---|---|
| | q = 0.25 | q = 0.5 | q = 0.75 | q = 0.25 | q = 0.5 | q = 0.75 |
| SATonGPU | n/a | 0.0171(7) | n/a | n/a | −5.9(3) | n/a |
| DAU | 0.0181(2) | 0.0185(4) | 0.0190(4) | −3.51(4) | −3.56(7) | −3.49(7) |
| SBM | 0.0211(7) | 0.0217(6) | 0.0234(8) | −2.6(2) | −2.6(1) | −2.7(1) |
| PT | 0.0239(1) | 0.0248(2) | 0.0252(1) | −0.92(2) | −0.97(4) | −0.97(2) |
| MEM | 0.030(9) | 0.025(2) | 0.024(3) | −1(2) | −0.6(2) | −0.2(2) |
| DWA | n/a | 0.08(4) | n/a | n/a | −6(2) | n/a |
| **p-bits: 2nd order (FPGA)** | **0.0194(1)** | **0.0206(2)** | **0.0210(3)** | **−3.70(4)** | **−3.76(6)** | **−3.74(2)** |
| **p-bits: 3rd order (FPGA)** | **0.0182(6)** | **0.0201(2)** | **0.0202(1)** | **−4.13(6)** | **−4.30(0)** | **−4.23(1)** |

We reproduce data from Table 2 of ref. [10] and add the corresponding numbers obtained for FPGA-based p-computers (highlighted in bold). Lower constants ($\gamma$, $\eta$) indicate better scaling and better prefactors, respectively.

Auxiliary spins extend the phase space exponentially, and transitions between viable solutions of clauses are hindered by additional barriers, energetic and entropic. Surprisingly, however, these "disadvantages" do not always lead to a scaling difference for an algorithm: the Adaptive Parallel Tempering we employ in this work shows identical scaling for the second and third-order formulation of the XORSAT problem with minor differences limited to prefactors. Thus, we conclude that even when an optimization problem is natively $k$-order, a $k$-local formulation may not lead to improved scaling. In the case of XORSAT, our results settle an open conjecture by Kowalsky et al.[10] regarding whether a third-order formulation would improve scaling exponents and may indicate an optimal scaling achieved by the APT algorithm. We hasten to add, however, that third-order interactions could still provide energy and area benefits due to the reduced number of spins required to represent a problem, even if they do not provide algorithmic scaling advantages.

We used a single FPGA unit that matches the prefactor and the scaling of dedicated and expensive ASICs. The scalability of our system could be improved significantly by the integration of larger or multiple FPGAs or using nanodevice-based p-computers, specifically, stochastic magnetic tunnel junctions (sMTJ)[19]. For such an sMTJ-based p-computer, detailed projections[19,32] indicate that a capacity of $N = 10^6$ p-bits on a single chip is feasible, given that magnetic memory technology has already integrated up to billions of MTJs in CMOS-compatible chips[33]. In addition, an MC sweep time of $\tau = 1$ ns is also feasible and has been experimentally demonstrated[34]. Having $N = 10^6$ p-bits would allow fitting a significant number of parallel runs (e.g., $f_p(n) \approx 480$ for $n = 208$ for the second order and $f_p(n) \approx 960$ for the third order assuming 10 replicas for all sizes). With the prefactor being a function of $n$, the slope slightly increases to 0.0218 and 0.0213, respectively.

The SAT-on-GPU approach used by ref. [17] shows the best TTS, both in terms of the prefactor and the scaling exponent. This shows the steep threshold of success for emerging Ising machines: they should not simply outperform each other but also be the best possible classical solvers. In this case, Gaussian elimination is not a relevant comparison because of how small modifications to the XORSAT problem can transform it into a much more general and harder $k$-SAT problem, rendering Gaussian elimination irrelevant. However, greedy algorithms and GPU-accelerated smart heuristics must be outperformed, taking into account all the preprocessing and data movement costs[35]. Despite these challenges, dedicated Ising solvers are steadily closing the gap. In fact, nanodevice implementations of p-bits could achieve orders of magnitude improvements over the current state-of-the-art, providing strong motivation for the engineering of these systems.

## Assumptions and qualifications

To maximize the transparency of our benchmarking approach, we collectively report all our assumptions and qualifications that have gone into producing our main result in Fig. 7. While we do not entirely ignore replica swap computations, as we explain below, we do not include replica exchange times in our calculations, which are currently done by an external CPU that communicates with the FPGA. This is similar to earlier versions of the DAU[5]. However, this is not a fundamental problem, and future versions can perform on-chip replica swaps with custom logic as the newer DAU does[10]. To estimate replica swap times, we consider replica exchanges as flipping all the spins in each replica. For each swap attempt, we add two full sweeps (each taking 66.67 ns) to our sweeps-to-solution calculations. This is a safe estimate as the on-chip replica swap can be performed at a much higher clock frequency, leading to lower swapping latencies. In such an implementation, the energy computation would be the costliest step, and we have achieved on-chip energy calculation latencies of approximately 56 ns for problem sizes up to $n = 256$. This is well below our two-sweep assumption. The only part that we have currently not considered is the probabilistic swap of $\beta$ values with a single random number, and this can be achieved within a clock cycle or two. Our fully on-chip PT implementation will be discussed elsewhere. We have not accounted for read-write times between FPGA and external CPU in our TTS calculations since the APT algorithm does not require rewriting the weights, which is a one-time cost. Even though the reconfigurable sparse master graph approach we introduce houses multiple instances of a given size in a single FPGA synthesis, it does require a new synthesis at different sizes. This is not a fundamental problem either since, for custom ASIC implementations (for example, with sMTJs or digital CMOS), much larger problem sizes can be housed on-chip. The FPGA data pertains to sizes ranging from $n = 16$ to $n = 112$ for second-order and $n = 160$ for third-order. Projections beyond these sizes are based on the CPU emulation in Fig. 3. The algorithmic sweeps to solution are the same in both FPGA and CPU implementations, indicating our scaling factors are robust. Due to the limitations of the moderate FPGAs we used, at $n = 96$ and $n = 112$, we were constrained to fitting only 50 instances into our second-order master graphs. However, we carefully checked that for the chosen 50 instances, the success probabilities were identical to those obtained from 100 instances. For third-order master graph implementation, owing to reduced hardware utilization, all 100 instances for all problem sizes till $n = 160$ were used for generating the FPGA data. In the implementation of the APT preprocessing algorithm for the third-order problems, we used exactly the same hyperparameters as those used for the second-order problems. This led to fewer replicas for third-order instances, which may have

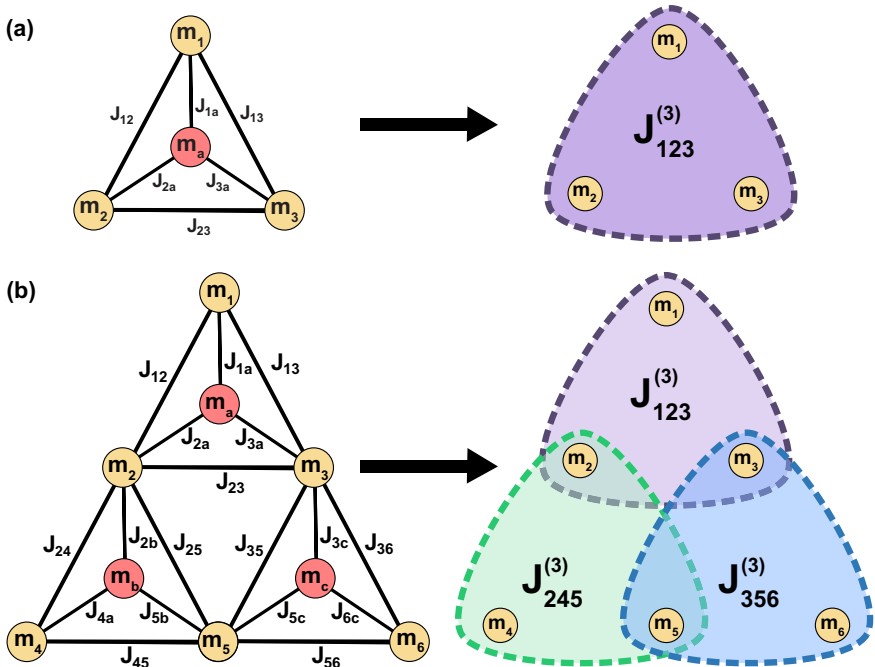

**Fig. 8 | Third-order instance generation. a** A single XORSAT gadget is reduced into cubic form. The auxiliary spin is removed, and the 6-edge weights are replaced by a single weight. **b** A 3-clause XORSAT problem is converted into cubic form. No spin is in more than three clauses.

deteriorated the time to solution (but not the scaling). Tuning the parameters of APT to match the number of replicas in the second order may lead to further performance advantages in optimized designs.

## Discussion

In this paper, we introduced a reconfigurable master graph architecture capable of solving multiple instances of sparse optimization problems. Our master graph modifies neighbor connectivity to reproduce all-to-all reconfigurability for sparse problems while maintaining a highly efficient and massively parallel Gibbs sampling procedure. This procedure is based on asynchronous (and colored) clocks for both second-order and third-order interaction-based p-computers. We implemented this architecture on FPGA-based probabilistic computers, demonstrating excellent algorithmic and prefactor performance in the XORSAT challenge, where several leading custom-built Ising machines have recently been evaluated. We also evaluated the performance of third-order p-computers for these problems, reporting a significant prefactor advantage without a corresponding improvement in the algorithmic exponent. Experimentally-informed projections suggest that nanodevice-based implementations of p-computers could achieve much greater improvements at larger scales.

A key limitation of the sparse master graph architecture we proposed is the requirement for the underlying instance to be relatively sparse. In sufficiently dense graphs, parallelism is effectively prohibited (as an all-to-all connected instance requires $\mathcal{O}(n)$ colors, reducing parallelism to serial updates). One solution is to sparsify dense problems using sparsification techniques[7].

Our motivation for using the XORSAT problem as a benchmark stems from the thorough evaluations of leading Ising machines on this problem. We believe that solving common combinatorial optimization (or sampling) problems with clearly articulated assumptions and qualifications will enable an accurate evaluation of Ising machines, highlighting both their strengths and weaknesses. The lack of a common suite of benchmarks has led to each Ising machine being evaluated on its own problem, making comparisons difficult.

## Methods

### Third-order instance generation

We follow the following procedure to convert second-order XORSAT problems to third-order (Fig. 8). For each auxiliary spin, we find the product of all of its weights and biases. If this number is positive, then that XORSAT clause is of the form

$$m_1 \oplus m_2 \oplus m_3 = 1$$

Otherwise, the clause equals −1. Then, for all non-auxiliary spins, we set the corresponding weight in the third-order tensor to ±1, equal to the literal to which the clause is equal. The resulting tensor is the weight tensor corresponding to the cubic form of the original XORSAT problem.

### Bipolar to binary weight conversion

As mentioned before, it is convenient to implement the MCMC algorithm in the FPGA in terms of binary variables, where $s_i \in \{0, 1\}$ and the random number lies in the interval [0, 1]. However, the weights for all orders of interactions need to be converted to their binary equivalents by following these equations after applying the standard map $m \rightarrow 2s − 1$ to bipolar variables $m$:

$$J_{ijk}^{'(3)} = 8 J_{ijk}^{(3)} \tag{9}$$

$$J_{ij}^{'(2)} = \sum_k \left( -4 J_{ijk}^{(3)} \right) + 4 J_{ij}^{(2)} \tag{10}$$

$$h_i' = \sum_{jk} J_{ijk}^{(3)} - 2 \sum_j J_{ij}^{(2)} + 2 h_i \tag{11}$$

where $J_{ijk}^{(3)}, J_{ij}^{(2)}$ and $h_i$ are the bipolar weights and biases, whereas $J_{ijk}^{'(3)}, J_{ij}^{'(2)}$ and $h_i'$ are the binary weights and biases. It is interesting to note that the conversion to binary introduces second-order and bias terms even for a purely third-order problem in the bipolar form. Eq. (2) is also converted to its corresponding binary

form:

$$s_i = \mathrm{sgn}\left[\theta(\beta I_i') - \mathrm{rand}_U(0,1)\right] \qquad (12)$$

Here, $s_i \in \{0, 1\}$, $\theta$ is the Heaviside function implemented using a 32-bit lookup table (LUT), $\mathrm{rand}_U(0, 1)$ is a 32-bit random number sampled from a uniform distribution between 0 and 1, generated using Xoshiro, and $I_i'$ is the binary synapse.

## Calculation of optimal median time to solution

For the calculation of TTS used in Eq. (7), we proceed exactly as described in ref. 10: once instance-wise success probabilities ($p_i(t_f)$) are estimated as a function of $t_f$, we calculate a $\mathrm{TTS}_i(t_f) = t_f(1 - 0.99)/(1 - p_i(t_f))$. Then, we bootstrap the median (or the other quartiles) of $\mathrm{TTS}_i(t_f)$ with 1000 samples. The mean of the bootstrapped distribution corresponds to $\langle \mathrm{TTS}(t_f)\rangle_q$, whose minimum is $\langle \mathrm{TTS}\rangle_q$.

Kowalsky et al.[10] calculate the $p_i(t_f)$ using a sophisticated approach; however, we adopt the same approach used in Fig. 2: $p_i(t_f)$ is calculated over 1000 runs where $t_f$ is measured up to a maximum of 3000 swap attempts. We found that both approaches produce exactly the same slope for p-computers and do not affect our comparisons.

## Data availability

All processed data generated in this study are provided in the main text. The data that supports the plots within this paper can be found in the GitHub repository[36]. Other findings of this study are available from the corresponding authors upon request.

## Code availability

The CPU codes running the Adaptive Parallel Tempering algorithm and our raw data used to produce the lower panel of Figs. 2, 3, and 7 can be found in the GitHub repository[36] mentioned in the data availability section.

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

## Acknowledgements

This work has been supported by an ONR-MURI grant N000142312708 and the Semiconductor Research Corporation (SRC). Use was made of computational facilities purchased with funds from the National Science Foundation (CNS-1725797) and administered by the Center for Scientific Computing (CSC). The CSC is supported by the California NanoSystems Institute and the Materials Research Science and Engineering Center (MRSEC; NSF DMR 2308708) at UC Santa Barbara. We gratefully acknowledge discussions with Andrea Grimaldi, Giovanni Finocchio, and Masoud Mohseni.

## Author contributions

S.N., S.K., N.A.A., and K.Y.C. conceived the study. K.Y.C. supervised the study. S.C., S.K., and S.N. developed the higher-order p-bit mapping, graph and hypergraph coloring, and algorithms used. S.N. and N.A.A. developed the all-to-all reconfigurable FPGA-based p-computer architecture. All authors have participated in discussing the results and helped improve the paper.

## Competing interests

The authors declare no competing interests.
