## [Peer Review file · Nature Communications]

All-to-all reconfigurability with sparse and higher-order Ising machines

Corresponding Author: Professor Kerem Camsari

Version 0:

Reviewer comments:

Reviewer #1

(Remarks to the Author)

The technical aspects of the work are sound, well-explained, and self-consistent. However, I do not find any flow in the data presented, although I consider the methodology sound.

Beyond the technical part, I have two major concerns regarding the content. Firstly, the benchmark used, 3R3X, does not offer a rich statistical picture, such as possible phase transitions, which are typical of other satisfiability benchmarks used in the literature. Additionally, its true hardness is not well-known but merely conjectured. The 3R3X is not even relevant as an “industrial” benchmark. It has recently gained some popularity due to a small community of people promoting it, but I do not see its real relevance for a journal like Nature Communications.

Secondly, while the technical aspects are sound, the novelty of the content is only incremental compared to the authors' previous works. They introduce sparsity to achieve some acceleration through parallelization. (Also, in my opinion, since the 3R3X is not fully connected, they should be able to leverage parallelization even in a fully connected p-bit system, but this goes beyond the review purpose). This is more of an implementation improvement rather than a novel scientific advancement.

Therefore, I believe this article would be more suitable for a more technical journal.

Reviewer #3

(Remarks to the Author)

The authors implement an Ising machine using FPGAs studying the 2-body and 3-body XORSAT problem using a variation of the parallel tempering algorithm. The main novel points of the work is the reconfigurability of the network architecture. The problems that they solve are all sparse problems, meaning that the interconnections between the spins only involve several couplings and the remaining are zero. Unlike other implementations where the couplings are fixed by hardware (e.g. in many quantum annealers qubits cannot be coupled beyond nearest neighbor), here any bit can be connected to any other bit, but the number of couplings cannot be large. By implementing 3-body connections they also implement the 3R3X problem. They also introduce an adaptive parallel tempering suitable for their architecture which involves some preprocessing that is problem dependent to determine suitable parallel tempering parameters. They provide a comprehensive documentation of their methods and their results, giving their final performance results in Fig. 7 where they compare against leading Ising machine implementations. They show that their methods are competitive and has potential towards even faster solvers with different hardware.

I think overall the paper is very well written and the authors have done a comprehensive job of analyzing their results. I would like to ask one thing which I feel I may not have understood adequately, and perhaps the authors would like to clarify. In Fig. 2 they show one of the central ideas that they are introducing in this paper, which is

the master graph. To me, this master graph doesn't seem to really help in simplifying the implementation, since when adding all the individual instances the master graph simply becomes a fully connected graph. This seems to say that potentially any bit can be connected to any other bit, so that there is no particular simplification. Or is there more to it than that? Perhaps the authors can explain this?

My other comment is that in their main results Fig. 7, from the way the authors plotted it, it appears that their results are on first glance the best in the field. But in fact the SAT-on-GPU results go from 250 to 600 bits and plotting a larger range would show that in fact these have a similar scaling with a smaller prefactor, so this is probably still the Ising machine to beat. The authors do say this clearly in the abstract which is excellent, but in the text there is one small mention. Perhaps discussing this a little more might be warranted.

In the conclusions the authors merely summarize the results of the paper and do not discuss future prospects such as the potentially applicability to other problem classes. Perhaps a little more discussion here would also be good.

But overall I think this is a fine paper and I would be happy to see this published in Nature Communications.

Minor comments/typos

- line 51: sovl
- line 381: strange font
- line 427: Silimar

Version 1:

Reviewer comments:

Reviewer #1

(Remarks to the Author)

The authors' responses to the reviewers' comments are largely unsatisfactory and appear to focus more on defending their position rather than addressing the core issues raised. This reviewer maintains that the benchmarks presented are neither relevant to industry nor academia. It is entirely feasible to refute the authors' points one by one, but it is likely that this would result in further reiterations of the same defense.

For instance, the fact that certain solvers have been tested on this benchmark does not, in itself, validate the benchmark's relevance. A significant portion of the testing seems to have been conducted by the same group that developed the benchmark, which raises concerns about objectivity. If the authors genuinely aim to "address a critical gap," as they claim, they would benefit from developing a far more robust and comprehensive set of benchmarks. They might look to examples such as MIPLIB and the extensive work conducted by the Zuse Institute Berlin in collaboration with the Integer Linear Programming (ILP) community. This is a well-recognized and serious effort to systematically benchmark ILP solvers, and it sets a far higher standard for what constitutes addressing a critical gap in the field.

Moreover, the relevance of solvers like Ising Machines to the XORSAT problem is questionable. There are numerous reasons for this. From an algorithmic perspective, Ising Machines are heuristics, and testing them on select samples provides little insight into the general XORSAT problem. The $O(n^3)$ complexity mentioned by the authors refers to a well-established algorithm with formally defined computational complexity. Even if Ising Machines were able to "crack" certain instances of the presented benchmarks numerically, this would still be irrelevant in terms of contributing to a meaningful complexity assessment for XORSAT. The authors' responses suggest that they may not fully grasp this distinction, possibly due to a lack of relevant background in computer science.

Finally, the authors' explanations regarding the novelty of their work remain unsatisfactory. Despite their detailed responses, the novelty appears to be incremental at best and more suited to a technical journal rather than a high-impact publication.

Reviewer #3

(Remarks to the Author)

I have read the rebuttal to my questions and the author referees, and the updates to the paper.

I believe the authors have satisfactorily addressed my concerns and now I am happy to recommend publication in Nature Communications.

Re: NCOMMS-24-30683

All-to-all reconfigurability with sparse and higher-order Ising machines

Srijan Nikhar,^{1,2} Sidharth Kannan,^{1,2} Navid Anjum Aadit,^{1,2} Shuvro Chowdhury,¹ and Kerem Y. Camsari¹

¹*Department of Electrical and Computer Engineering,
University of California, Santa Barbara, Santa Barbara, CA, 93106, USA*

²*Equally contributing authors*

(Dated: August 12, 2024)

I. REVIEWER 1

- **#1.0** *The technical aspects of the work are sound, well-explained, and self-consistent. However, I do not find any flow in the data presented, although I consider the methodology sound.*

AUTHOR RESPONSE

We appreciate comments and we are grateful for the positive comments about the technical aspects and the methodology of the paper. We also appreciate the thoughtful suggestion to make the data flow more consistent and have revised the main manuscript accordingly. Please see our point-by-point responses (author response) and accompanying changes (author action) below. We have also attached a marked up manuscript that highlights the changes.

- **#1.1** *Beyond the technical part, I have two major concerns regarding the content. Firstly, the benchmark used, 3R3X, does not offer a rich statistical picture, such as possible phase transitions, which are typical of other satisfiability benchmarks used in the literature. Additionally, its true hardness is not well-known but merely conjectured. The 3R3X is not even relevant as an “industrial” benchmark. It has recently gained some popularity due to a small community of people promoting it, but I do not see its real relevance for a journal like Nature Communications.*

AUTHOR RESPONSE

The reviewer argues that because the Exclusive OR Satisfiability (XORSAT) problem is not an industrial benchmark with real-world relevance or that it has gained some popularity due to a small community, it is not relevant for Nature Communications.

We believe that the relevance of the XORSAT problem comes *precisely from this small but steadily growing community* of researchers from industry and academia who have used this problem to benchmark their Ising machines. We believe that the united effort in benchmarking Ising machines around a common set of problems *addresses a critical gap* in the Ising machine community where every solver is *typically evaluated on a different problem!*

Indeed, the XORSAT challenge, as proposed, may be the only combinatorial optimization problem that concretely compares various leading Ising Machine (IM) implementations such as Fujitsu’s Digital Annealer, Toshiba’s Simulated Bifurcation Machine, UCSD’s Memcomputing, D-Wave’s quantum annealers as well as specialized heuristic solvers running on GPUs. Our purpose in this manuscript is to add p-bits to this growing list to have a common platform that allows a concrete evaluation of emerging IMs. *In this sense, we think Nature Communications is precisely the right type of venue for this work* to encourage a broad range of researchers to benchmark their Ising machines on a common problem.

The reviewer’s comments about the true hardness of the XORSAT problem do not seem to be correct: the computational complexity of the 3-regular 3-XORSAT problem is very well-understood: XORSAT has polynomial time ($\mathcal{O}(N^3)$) complexity in a deterministic computer by Gaussian elimination and its decision version firmly belongs in the complexity class P. The remarkable feature of this problem is that despite being in P, it poses great difficulties to the category of stochastic local search solvers Ising machines belong to. Virtually all Ising machines exhibit exponential scaling for time to solution (in the problem size) for XORSAT, which makes it a fascinating example of how different formulations of a combinatorial optimization problem

may render them easy or hard.

Regarding the “richness” of XORSAT, it is important to note that the XORSAT problem has long played an important role in theoretical computer science, especially regarding the $P \stackrel{?}{=} NP$ question [1], since any proof that relates computational hardness to glassy energy landscapes must explain why XORSAT is not hard while 3SAT is (see, “Being glassy without being hard to solve” [2].) In this regard, if Ising machines “crack” the XORSAT problem, this may have far-reaching implications in computation.

Finally, we note that any major improvement in the XORSAT problem will likely have practical implications for other types of SAT problems which have very similar mappings as far as the Ising machines are concerned. Moreover, establishing a common benchmark is essential for the progress of the field, and Nature Communications, with its broad reach, is the ideal venue to encourage this effort. In conclusion, we believe that the wide relevance of the XORSAT challenge comes not from its industrial or practical relevance but from the fact that it has been benchmarked by a large subset of the Ising community. We believe the addition of p-bits to this list improves its relevance.

AUTHOR ACTION

Please see the revised Section I (Introduction) and the Conclusion where we discuss the broad relevance of the XORSAT problem, in the context of Ising machine research that lacks clear benchmarks.

- **#1.2** *Secondly, while the technical aspects are sound, the novelty of the content is only incremental compared to the authors’ previous works. They introduce sparsity to achieve some acceleration through parallelization. (Also, in my opinion, since the 3R3X is not fully connected, they should be able to leverage parallelization even in a fully connected p-bit system, but this goes beyond the review purpose). This is more of an implementation improvement rather than a novel scientific advancement.*

Therefore, I believe this article would be more suitable for a more technical journal.

AUTHOR RESPONSE

We appreciate the positive comments regarding technical soundness. We would like to clarify however that our manuscript achieves **two key significant results** beyond a simple implementation of previous ideas. The high-performance we have obtained in this work with FPGA-based p-bits would not be possible without these two novelties.

The first is a ***new type of master-graph approach that allows highly-parallel (chromatic) Gibbs sampling*** without losing reconfigurability. As we explain below, this is fundamentally different from a static graph with all-to-all connectivity. The second is the ***first implementation of a third order Ising machine applied to a large scale problem***. The third order interaction extends J_{ij}, h_i type interactions to include third order tensors J_{ijk} , ***in hardware***. We elaborate on our key contributions below:

- **Key novelty #1: Master graph:** One of the problems of Ising machines is their lack of reconfigurability to encode different problems. This difficulty is usually circumvented by employing all-to-all graphs that allow connections between all nodes. This approach however is not straightforwardly scalable or parallelizable. The master graph-based approach introduced in this paper, on the other hand, does exploit parallelism, and is fundamentally different from a typical all-to-all graph configuration. In an all-to-all graph, each spin needs to be updated sequentially, because all connections *can be activated* in principle. A single spin, for example, may have of a number of neighbors of order $\mathcal{O}(n)$, as such, there is no obvious mechanism for graph coloring or block updating in these static, all-to-all networks.

In our sparse master-graph approach, however, before the graph is loaded to our machine, the graph is colored using a quick heuristic polynomial time algorithm. This step splits the graph into pieces that can be updated in parallel. Then, through careful multiplexing, we assign polychronous (out-of-phase and synchronous) clocks for a given node. Note that in this scheme, a node can be (and does get) assigned a different clock (color) depending on the instance. This approach allows us to update the entire network in just a few blocks, enabling massive parallelization. As such, our new design can handle all instances of the XORSAT problem in a single synthesis while preserving sparsity by switching between instances using clock and neighbor multiplexers. Importantly, our architectural improvement is not limited to FPGA implementations as it can also be applied to custom ASICs, with broad applicability.

To summarize, there are concrete and measurable differences between an all-to-all network and our sparse master-graph approach:

1. **Sequential vs Parallel Update:** An all-to-all system requires $\mathcal{O}(N)$ colors to be colored, effectively becoming a sequentially updating machine. In contrast, the sparse formulations of the XORSAT problem (or other SAT problems) allow us to use $\mathcal{O}(1)$ colors, and this leads to an architectural scaling improvement as we demonstrated in FIG. 3 and FIG. 6.
 2. **Scalability:** In an all-to-all network, the adder complexity that performs the synaptic summation grows as $\mathcal{O}(N^2)$ due to the $N^2 - N$ connections. In contrast, our sparse network maintains a k -local ($k \ll N$) structure through neighbor multiplexing, resulting in an adder complexity of $\mathcal{O}(kN)$. Consequently, the synapse equation $I_i = \sum_j (J_{ij}m_j + h_i)$ can be computed much faster, allowing the sampling of the network at much higher speeds.
- **Key novelty #2: Higher-order Ising machine for a large-scale problem:** The use of higher-order interactions in the study of Ising machines have received significant attention [3–10]. In this work, we have taken a concrete step towards evaluating higher-order Ising machines and implemented, to the best of our knowledge, the first-ever hardware realization of a higher-order Ising machine to solve a large scale problem. In doing so, we were able to settle an open conjecture posed by Kowalsky *et al.* that questioned whether using higher-order interactions in the solution of the XORSAT problem changes the algorithmic complexity of the Ising machine. We have found that while third-order interactions required only half of the spins second-order requires to represent an XORSAT problem and provided significant prefactor improvements as a result, third-order interactions **did NOT change** the algorithmic scaling of our solver, which had a similar slope to many other solvers. That said, we concluded that the third order representation could still lead to area and energy benefits which are relevant for Ising machines. These are significant new results, relevant in the context of evaluating the potential of higher-order interactions for Ising machines.

AUTHOR ACTION

Please see Section II C where we clarified how the master graph approach differs from an ordinary all-to-all network implementation.

II. REVIEWER 3

- **#3.0** *The authors implement an Ising machine using FPGAs studying the 2-body and 3-body XORSAT problem using a variation of the parallel tempering algorithm. The main novel points of the work is the reconfigurability of the network architecture. The problems that they solve are all sparse problems, meaning that the interconnections between the spins only involve several couplings and the remaining are zero. Unlike other implementations where the couplings are fixed by hardware (e.g. in many quantum annealers qubits cannot be coupled beyond nearest neighbor), here any bit can be connected to any other bit, but the number of couplings cannot be large. By implementing 3-body connections they also implement the 3R3X problem. They also introduce an adaptive parallel tempering suitable for their architecture which involves some preprocessing that is problem dependent to determine suitable parallel tempering parameters. They provide a comprehensive documentation of their methods and their results, giving their final performance results in Fig. 7 where they compare against leading Ising machine implementations. They show that their methods are competitive and has potential towards even faster solvers with different hardware.*

AUTHOR RESPONSE

We thank the reviewer for the detailed review including the critical highlights of the paper and for providing encouraging comments and useful suggestions. Below, we provide a point-by-point response to the reviewer comments (author response), followed by changes in the manuscript (author action). We have also attached a marked up manuscript that highlights the changes.

- **#3.1** *I think overall the paper is very well written and the authors have done a comprehensive job of analyzing their results. I would like to ask one thing which I feel I may not have understood adequately, and perhaps the authors would like to clarify. In Fig. 2 they show one of the central ideas that they are introducing in this paper, which is the master graph. To me, this master graph doesn't seem to really help in simplifying the implementation, since when adding all the individual instances the master graph simply becomes a fully connected graph. This seems to say that potentially any bit can be connected to any other bit, so that there is no particular simplification. Or is there more to it than that? Perhaps the authors can explain this?*

AUTHOR RESPONSE

We appreciate the reviewer’s insightful question and agree that further clarification of the central ideas was necessary. Indeed, we believe there is “more” to our approach than a simple all-to-all graph with static connectivity, as we elaborate below.

- One of the problems of Ising machines is their lack of reconfigurability to encode different problems. This difficulty is usually circumvented by employing all-to-all graphs that allow connections between all nodes. This approach however is not straightforwardly scalable or parallelizable. The master graph-based approach introduced in this paper, on the other hand, does exploit parallelism, and is fundamentally different from a typical all-to-all graph configuration. In an all-to-all graph, each spin needs to be updated sequentially, because all connections *can be activated* in principle. A single spin, for example, may have of a number of neighbors of order $\mathcal{O}(n)$, as such, there is no obvious mechanism for graph coloring or block updating in these static, all-to-all networks

In our sparse master-graph approach, however, before the graph is loaded to our machine, the graph is colored using a quick heuristic polynomial time algorithm. This step splits the graph into pieces that can be updated in parallel. Then, through careful multiplexing, we assign polychronous (out-of-phase and synchronous) clocks for a given node. Note that a node can be (and does get) assigned a different clock (color) depending on the instance. This approach allows us to update the entire network in just a few blocks, enabling massive parallelization. As such, our new design can handle all instances of the XORSAT problem in a single synthesis while preserving sparsity by switching between instances using clock and neighbor multiplexers. Importantly, our architecture improvement is not limited to FPGA implementations as it can also be applied to custom ASICs, with broad applicability.

To summarize, there are concrete and measurable differences between an all-to-all network and our sparse master-graph approach:

1. **Sequential vs Parallel Update:** An all-to-all system requires $\mathcal{O}(N)$ colors to be colored, effectively becoming a sequentially updating machine. In contrast, the sparse formulations of the XORSAT problem (or other SAT problems) allow us to use $\mathcal{O}(1)$ colors, and this leads to an architectural scaling improvement as we demonstrated in FIG. 3 and FIG. 6..
2. **Scalability:** In an all-to-all network, the adder complexity that performs the synaptic summation grows as $\mathcal{O}(N^2)$ due to the $N^2 - N$ connections. In contrast, our sparse network maintains a k -local ($k \ll N$) structure through neighbor multiplexing, resulting in an adder complexity of $\mathcal{O}(kN)$. Consequently, the synapse equation $I_i = \sum_j (J_{ij}m_j + h_i)$ can be computed much faster, allowing the sampling of the network at much higher speeds.

We agree with the reviewer that the design is not simple and still pays the price for accommodating all the instances. However, it is important to emphasize that our approach exploits the *maximum amount of parallelism* that can be extracted from a sparse instance, which is not nominally possible in a static all-to-all network. This leads us to conclude that if dense problem graphs can be *sparsified*, they can be operated much faster. We have elaborated on these ideas in the revised manuscript and in the conclusion section where we have included future directions.

AUTHOR ACTION

Please see the revised Section II C where we clarify the differences between a static all-to-all network and our master-graph architecture. Please see the revised section III (Conclusion) with some future directions.

- **#3.2** *My other comment is that in their main results Fig. 7, from the way the authors plotted it, it appears that their results are on first glance the best in the field. But in fact the SAT-on-GPU results go from 250 to 600 bits and plotting a larger range would show that in fact these have a similar scaling with a smaller prefactor, so this is probably still the Ising machine to beat. The authors do say this clearly in the abstract which is excellent, but in the text there is one small mention. Perhaps discussing this a little more might be warranted.*

AUTHOR RESPONSE

We agree with the reviewer’s comments regarding FIG. 7 and the scaling behavior of the SAT-on-GPU results. We acknowledge that the SAT-on-GPU algorithm exhibits excellent scaling and a smaller prefactor compared to other methods as mentioned in our abstract and noted by the reviewer.

We understand the importance of providing a complete comparison and, as suggested, we have extended our figures to include problem sizes up to 640 bits to fully capture the performance of SAT-on-GPU. We have also expanded the discussion in the text

Fig. 7. Optimal median time to solution (TTS) for quartile $q = 0.5$ is shown as a function of problem size, n , for different state-of-the-art solvers: we adapt the figure from [10] for other solvers and add the results from the p-computer onto it. The solid red stars represent measured data for the FPGA-based p-computer. Hollow red squares indicate FPGA projections based on the CPU data from Figure 3. Error bars obtained from 95% confidence intervals are smaller than the size of the markers and omitted. The green rectangles represent projections for stochastic Magnetic Tunnel Junction (sMTJ)-based p-computers assuming 10 to 50 replicas fitted in a monolithic chip with 1 million sMTJs, where each sMTJ is assumed to fluctuate with $\tau = 1$ ns, yielding 1 million flips per nanosecond for the entire network and 1 ns of sweep time at all sizes. (see the text).

to explain how SAT-on-GPU constitutes the best classical approach, and the real target to outperform for Ising machines.

In the modified figure where we now show the SAT-on-GPU on a larger scale, we also showed nanodevice projections, prominently labeling these as mere projections, however.

AUTHOR ACTION

Please see the revised FIG. 7 (reproduced here) and the related discussion in Section III F.

- **#3.3** *In the conclusions the authors merely summarize the results of the paper and do not discuss future prospects such as the potentially applicability to other problem classes. Perhaps a little more discussion here would also be good.*

AUTHOR RESPONSE/ACTION

We agree. We have now extended the conclusion (Section III) with the potential applicability of our architecture to other problem classes, the significance of sparsity and how it can enable massive-parallelism even for nominally dense problems.

- **#3.4** *But overall I think this is a fine paper and I would be happy to see this published in Nature Communications.*

AUTHOR RESPONSE

We appreciate the thorough review and the positive comments.

- **#3.5** *Minor comments/typos*
 - line 51: sovl
 - line 381: strange font
 - line 427: Silimar

AUTHOR RESPONSE/ACTION

We thank the reviewer for the careful reading. We have addressed these in the revised manuscript.

REFERENCES

- [1] MIT-News, 3 questions: P vs. NP (2010), accessed: 2024-08-10.
- [2] F. Ricci-Tersenghi, Being glassy without being hard to solve, *Science* **330**, 1639 (2010).
- [3] W. A. Borders, A. Z. Pervaiz, S. Fukami, K. Y. Camsari, H. Ohno, and S. Datta, Integer factorization using stochastic magnetic tunnel junctions, *Nature* (2019).
- [4] Y. He, C. Fang, S. Luo, and G. Liang, Many-body effects-based invertible logic with a simple energy landscape and high accuracy, *IEEE Journal on Exploratory Solid-State Computational Devices and Circuits* (2023).
- [5] C. Bybee, D. Kleyko, D. E. Nikonov, A. Khosrowshahi, B. A. Olshausen, and F. T. Sommer, Efficient optimization with higher-order ising machines, arXiv preprint arXiv:2212.03426 (2022).
- [6] T. Kanao and H. Goto, Simulated bifurcation for higher-order cost functions, *Applied Physics Express* **16**, 014501 (2022).
- [7] M. Hizzani, A. Heitmann, G. Hutchinson, D. Dobrynin, T. V. Vaerenbergh, T. Bhattacharya, A. Renaudineau, D. Strukov, and J. P. Strachan, Memristor-based hardware and algorithms for higher-order hopfield optimization solver outperforming quadratic ising machines (2023), arXiv:2311.01171 [cs.ET].
- [8] Y. Su, T. T.-H. Kim, and B. Kim, A reconfigurable cmos ising machine with three-body spin interactions for solving boolean satisfiability with direct mapping, *IEEE Solid-State Circuits Letters* **6**, 221 (2023).
- [9] Y. He, C. Fang, S. Luo, and G. Liang, Logically synthesized invertible logic based on many-body effects with probabilistic-bit implementation, in *2023 Silicon Nanoelectronics Workshop (SNW)* (2023) pp. 39–40.
- [10] T. Bhattacharya, G. H. Hutchinson, G. Pedretti, X. Sheng, J. Ignowski, T. V. Vaerenbergh, R. Beausoleil, J. P. Strachan, and D. B. Strukov, Computing high-degree polynomial gradients in memory (2024), arXiv:2401.16204 [cs.ET].

Re: NCOMMS-24-30683A

All-to-all reconfigurability with sparse and higher-order Ising machines

Srijan Nikhar,^{1,2} Sidharth Kannan,^{1,2} Navid Anjum Aadit,^{1,2} Shuvro Chowdhury,¹ and Kerem Y. Camsari¹

¹Department of Electrical and Computer Engineering,
University of California, Santa Barbara, Santa Barbara, CA, 93106, USA

²Equally contributing authors

(Dated: September 26, 2024)

I. REVIEWER 3

- **#3.0** *I have read the rebuttal to my questions and the aother referees, and the updates to the paper. I believe the authors have satisfactorily addressed my concerns and now I am happy to recommend publication in Nature Communications.*

AUTHOR RESPONSE

We appreciate the reviewer's thoughtful comments and we are delighted to hear the reviewer's concerns have been satisfactorily addressed.

II. REVIEWER 1

- **#1.0** *The authors' responses to the reviewers' comments are largely unsatisfactory and appear to focus more on defending their position rather than addressing the core issues raised. This reviewer maintains that the benchmarks presented are neither relevant to industry nor academia. It is entirely feasible to refute the authors' points one by one, but it is likely that this would result in further reiterations of the same defense.*

AUTHOR RESPONSE

We appreciate the reviewer giving the revised manuscript another look. We are puzzled by the strong statement by the reviewer about the relevance of our benchmarks, especially since ours is a *follow-up* study to an existing benchmark on which *many other* Ising machines have been evaluated (Toshiba, Fujitsu, D-Wave, memcomputing, heuristic algorithms in CPUs and GPUs). The problem is clearly of relevance to *some* practitioners, in academia and industry.

The reviewer then remarks that it is "entirely feasible to refute authors' points one by one" but unfortunately does not do so. Please see our responses to any remaining concerns below.

- **#1.1** *For instance, the fact that certain solvers have been tested on this benchmark does not, in itself, validate the benchmark's relevance. A significant portion of the testing seems to have been conducted by the same group that developed the benchmark, which raises concerns about objectivity. If the authors genuinely aim to "address a critical gap," as they claim, they would benefit from developing a far more robust and comprehensive set of benchmarks. They might look to examples such as MIPLIB and the extensive work conducted by the Zuse Institute Berlin in collaboration with the Integer Linear Programming (ILP) community. This is a well-recognized and serious effort to systematically benchmark ILP solvers, and it sets a far higher standard for what constitutes addressing a critical gap in the field.*

AUTHOR RESPONSE

We understand there are many other possible problems to be solved by Ising Machines and ILP is another noteworthy target. We do not believe, however, that the lack of our focus to the particular community (or the problems) the reviewer favors diminishes the contributions of our study.

We chose the XORSAT problem *particularly because* it was empirically benchmarked against leading solvers in the growing Ising machine community, from academia and industry alike. Among hard optimization problems, XORSAT seems generic for Ising machines, exhibiting exponential complexity, despite not being NP-hard. In this sense, the XORSAT problem is as good as any other benchmark to draw hardware, architecture and algorithms related comparisons.

Among other contributions that we described in detail in our response in the previous round, one of our *problem-independent* contributions is to solve the reconfigurability problem of Ising machines (at least for sparse problems) while holding to massively parallel, chromatic Gibbs sampling *in hardware*, leading to an $\mathcal{O}(n)$ scaling advantage, which is universally useful.

- **#1.2** *Moreover, the relevance of solvers like Ising Machines to the XORSAT problem is questionable. There are numerous reasons for this. From an algorithmic perspective, Ising Machines are heuristics, and testing them on select samples provides little insight into the general XORSAT problem. The $\mathcal{O}(n^3)$ complexity mentioned by the authors refers to a well-established algorithm with formally defined computational complexity. Even if Ising Machines were able to “crack” certain instances of the presented benchmarks numerically, this would still be irrelevant in terms of contributing to a meaningful complexity assessment for XORSAT. The authors’ responses suggest that they may not fully grasp this distinction, possibly due to a lack of relevant background in computer science.*

AUTHOR RESPONSE

The $\mathcal{O}(n^3)$ comment we made was intended to remove any confusion about the hardness of the XORSAT problem, as a response to an earlier comment by the reviewer. While XORSAT is solvable in polynomial time and not NP-hard, this is not central to our contributions. Our work is about how a single FPGA implementation of our graph-colored architecture shows competitive performance compared to leading Ising solvers (while solving the reconfigurability problem in sparse networks). Our architecture would work for any other QUBO *and* PUBO formulation, where generalizing quadratic interactions to polynomial ones only incur logarithmically scaling hardware costs. These contributions make specifics about the XORSAT problem less important as they are universally applicable.

Finally, effectively dealing with entropic barriers in hard combinatorial optimization by intelligent algorithms is of fundamental interest (see, for example, arxiv.org/abs/2102.00182 or arxiv.org/abs/2111.13628), even when theoretically establishing worst-case lower bounds may be challenging or impossible. As such, our objective in this paper has been to empirically investigate the XORSAT problem as a generic, hard optimization problem to benchmark our reconfigurable and parallelized architecture against other Ising machines.